# Overlap of Suspicious and Non-Suspicious Features in the Ultrasound Evaluations of Leiomyosarcoma: A Single-Center Experience

**DOI:** 10.3390/diagnostics13030543

**Published:** 2023-02-02

**Authors:** Francesca Arezzo, Gennaro Cormio, Carmela Putino, Nicola Di Lillo, Erica Silvestris, Anila Kardhashi, Ambrogio Cazzolla, Claudio Lombardi, Michele Mongelli, Gerardo Cazzato, Vera Loizzi

**Affiliations:** 1Gynecologic Oncology Unit, IRCCS Istituto Tumori “Giovanni Paolo II”, 70124 Bari, Italy; 2Department of Biomedical Sciences and Human Oncology, University of Bari “Aldo Moro”, 70124 Bari, Italy; 3Interdisciplinar Department of Medicine, University of Bari “Aldo Moro”, 70124 Bari, Italy; 4Section of Molecular Pathology, Department of Emergency and Organ Transplantation, University of Bari “Aldo Moro”, 70124 Bari, Italy

**Keywords:** gynecological ultrasound, leiomyosarcoma, myometrial tumors

## Abstract

Leiomyosarcoma (LMS) is a rare type of mesenchymal tumor. Suspecting LMS before surgery is crucial for proper patient management. Ultrasound is the primary method for assessing myometrial lesions. The overlapping of clinical, laboratory, as well as ultrasound features between fibroids and LMS makes differential diagnosis difficult. We report our single-center experience in ultrasound imaging assessment of LMS patients, highlighting that misleading findings such as shadowing and absent or minimal vascularization may also occur in LMS. To avoid mistakes, a comprehensive evaluation of potentially overlapping ultrasound features is necessary in preoperative ultrasound evaluations of all myometrial tumors.

## 1. Introduction

Approximately 40–80% of women may develop leiomyomas, the most common benign gynecological disease, during their lifetime [1]. Uterine sarcoma, on the other hand, is a rare disease with an incidence ranging from 1.55 to 1.95 per 100,000 women per year [2]. According to the World Health Organization (WHO) in 2011, a leiomyosarcoma (LMS) is a specific type of rare sarcoma that accounts for over 60% of all cases of uterine sarcoma [3].

It is important to correctly evaluate myometrial tumors before surgery to ensure proper patient management and avoid delayed treatment. However, distinguishing an LMS from fibroids can be difficult due to similar clinical, laboratory, and ultrasound features [4,5,6].

Ultrasound is a cost-effective, non-invasive, and widely accepted imaging method for evaluating the myometrium. The reporting of ultrasound characteristics of the myometrium and myometrial lesions has been standardized through the development of a consensus statement by the Morphological Uterus Sonographic Assessment (MUSA) group [7].

The MUSA consensus statement advises evaluating certain predefined ultrasound characteristics, specifically the largest diameter of the myometrial tumor (mm), the number of lesions (single/multiple), the echogenicity of the solid tissue of the tumor (homogeneous/in-homogeneous), the tumor border (regular/irregular), the presence of cystic areas (yes/no), the presence of shadows (yes/no), subjective color score (1/2/3/4), and the vascular pattern of a myometrial lesion (circumferential/intralesional) [7].

On ultrasound, uterine fibroids typically appear as multiple, distinct, round lesions within the myometrium. They may display shadows at the edge of the lesion and/or internal fan-shaped shadowing. The echogenicity of fibroids can vary depending on the various components within them, such as muscle cells, fibrous stroma, calcification, and lipomatous or hyaline degeneration [8]. Fibroids often display surrounding blood flow visibly on ultrasound. The blood vessels in the surrounding area of fibroids tend to be more prominent than those in the normal myometrium [9].

On the contrary, an LMS generally appears on ultrasound as a single solid mass with a large diameter and inhomogeneous echogenicity of the solid tissue. Sometimes the masses contain cystic areas, typically irregular, with soft surfaces. Shadowing or calcifications are usually absent [10].

Therefore, it is important to assess multiple ultrasound features established by the MUSA group when evaluating myometrial tumors on ultrasound.

Hence, we aimed to describe the overlapping ultrasound features between fibroids and malignant myometrial lesions.

## 2. Methods

Here we report our single-center experience in ultrasound imaging assessment in LMS patients referred to our tertiary care center between May 2020 and May 2022.

During this observation period, 1000 patients underwent surgery for myometrial tumors. All patients received a preoperative transvaginal or trans-rectal ultrasound examination and additional trans-abdominal ultrasound when necessary. Ultrasound examinations were performed by a specialized ultrasound examiner with a 5.0–9.0 MHz vaginal probe or 3.5–5.0 MHz abdominal probe. All ultrasound reports and images were available for analysis.

Any cases that were uncertain or suspected to be LMS underwent laparotomy.

During the observation period, we evaluated 7 patients with LMS.

All information from the original ultrasound reports was collected based on ultrasound features according to the MUSA consensus statement [7]. We also evaluated the consistency of the lesion as perceived by probe pressure, reported as hard or soft.

The study was conducted according to the guidelines of the Declaration of Helsinki and approved by the Ethics Committee of the Azienda Ospedaliera Policlinico Consorziale-University of Bari, Italy (protocol code 6398).

## 3. Results

In our series, patients had a mean age ± sd of 50.2 ± 7.0. The mean largest diameter of LMS was 152.2 ± 91 mm. Five patients (71.4%) had a single myometrial lesion.

All LMS ultrasound images showed inhomogeneous echogenicity of solid tissue, intralesional vascularization, and soft consistency of the lesion at probe pressure.

All cases except one (85.7%) had irregular tumor borders, and all except one (85.7%) had cystic areas.

We found that four images (57.1%) had fan-shaped shadowing, and four pictures (57.1%) had minimal vascularization (CS2) (Table 1 and Figure 1).

## 4. Discussion

Unfortunately, LMS does not have any specific symptoms associated with it. Patients may experience pelvic pain, abnormal uterine bleeding, or a palpable pelvic mass, but these symptoms are also common in patients with uterine fibroids [5].

In 2007, Exacoustos et al. published a series on LMS, which included eight patients. The LMS examples reported were larger in size, with 88% having a diameter greater than 8 cm. They also had inhomogeneous echostructures, with degenerative cystic areas present in 50% of cases, and intense central vascularization [11].

In a study by Bonneau et al., 23 malignant myometrial lesions were described, including 3 LMS cases. The other cases were rhabdomyosarcoma, endometrial stromal sarcomas, undifferentiated endometrial sarcomas, stromal tumor of uncertain malignant potential (STUMP), and carcinosarcomas. Compared to benign lesions, malignant myometrial tumors appeared as single masses, without acoustic shadowing, and with a non-myometrial origin [12].

Testa et al., reported that malignant myometrial lesions can be characterized by large diameter, inhomogeneous echostructure, irregular anechoic areas, and absence of “radial stripy echogenicity” [8].

Given the complexity of the diagnosis, it is important to evaluate all individual clinical characteristics and ultrasound features to try to distinguish between benign and malignant myometrial tumors.

The rapid *growth* of a myometrial lesion may be suspected. However, the meaning of “rapid growth” is not well defined [13,14]. Additionally, studies have shown that even benign lesions can increase their size by 18% to 120% per year [15]. It is impossible to compare the growth of fibroids and LMS, as the majority of LMS cases are diagnosed after surgery.

Considering the age at diagnosis, some studies have reported that uterine fibroids are diagnosed at a younger age, with a mean age ranging from 40 to 51.7 [16,17]. However, they are also found in younger age groups, with some rare cases reported in adolescence [18]. On the other hand, LMS are generally diagnosed after menopause, with a mean age range from 44.6 to 58.1 [19,20]. According to previous studies, the mean age *±* sd of patients in our series was 50.2 ± 7.0 years and the median (sd) was 49.5 (7.0).

Considering laboratory tests, studies that support the use of *carbohydrate antigen 125 (CA125)* to differentiate LMS preoperatively from fibroids are conflicting.

Some authors suggest that elevated levels of CA125 are indicative of LMS [21], others associate high CA125 levels with advanced stages of the disease [20], and others deny any correlation [22]. As a result, relying solely on CA125 levels seems to be ineffective. In our series, CA125 levels were not indicative.

Another laboratory test that has been studied to suspect the diagnosis of LMS preoperatively involves *lactate dehydrogenase (LDH).* The first study on this topic was conducted by Seki et al. in 1992, in which they found that serum LDH levels were abnormally elevated in three out of seven (42.8%) patients with LMS [23]. In 2015, a study found that serum LDH levels greater or equal to 279 U/L were observed in 7 out of 15 patients (46.7%) with uterine sarcomas [24]. However, LDH is not a specific indicator of LMS and can be present in different types of cancer, such as hepatocellular carcinoma and breast cancer, as well as in uterine fibroids and degenerated uterine fibroids [25,26]. Therefore, LDH does not seem to help differentiate benign from malignant myometrial tumors [5].

Therefore, it is crucial to have an accurate imaging method to preoperatively identify suspected cases. Ultrasound is the primary method for assessing myometrial tumors, and this evaluation relies on the examination of several features established by the MUSA group.

Considering the *maximum diameter of the lesion,* a number of studies have shown that myometrial tumors with a diameter greater than 8 cm may raise suspicion of malignant disease [27]. In a study by Ludovisi et al., the median largest diameter of LMS was 10.6 cm [10]. Similarly, the mean diameter of LMS in a study by Chen et al. was 9.0 cm (±5.9 cm), while the mean diameter of fibroids was 8.5 cm (±3.9 cm) [28].

These studies suggest that LMS and fibroids may not differ greatly in terms of largest diameter. However, our findings indicate that LMS at the time of diagnosis had a significantly larger size than previously reported, with a mean largest diameter of 152.2 (±91) mm and a median (sd) of 93.5 (91).

*Internal shadows* and *fan-shaped shadowing* generally are typically seen in fibroids or adenomyosis and are not commonly reported in uterine sarcomas. However, a study by Bonneau et al. found that 20% of sarcomas had internal shadows, while only 2% had fan-shaped shadowing [12]. Ludovisi et al. reported that, after reviewing the images, a panel of experts found that 36% of images of uterine sarcoma had internal shadows or fan-shaped shadowing [10]. Similarly, in our experience, we found that, although shadows are generally indicative of benign tumors, four (57.1%) LMS images had shadows. This suggests that the presence of shadows during an ultrasound evaluation of a myometrial formation does not necessarily indicate a benign tumor.

Regarding *vascularization*, typically an LMS may have moderate to high vascularization, with blood vessels concentrated in the center of the lesion as evaluated by power Doppler sonography [11]. However, Ludovisi et al. found that about a quarter of the LMS in their study had minimal or absent vascularization [10]. In the analysis of highly vascularized mesenchymal masses reported by Russo et al., only 7% of highly vascularized myometrial tumors were malignant, and all of them were in women over 40 years old [29]. Similarly, our experience shows that vascularization is not a conclusive factor. In our series, four LMS images (57.1%) had minimal vascularization.

At the same time, evaluating a benign mesenchymal tumor can often be challenging due to the presence of concerning ultrasound features or the lack of typical ultrasound characteristics. Additionally, degenerations of uterine fibroids and histologic variants of benign leiomyomas make this assessment even more difficult.

In the series of Russo et al., in uterine myometrial lesions with high vascularization on ultrasound, the histological diagnosis was fibroids in 93% of cases, including leiomyoma variants in 45% and adenomyomas in 6%. Thus, the vascularization feature can be misleading.

Typically, fibroids have homogeneous echogenicity on ultrasound, but in the study of Russo et al., 72.3% of benign lesions had inhomogeneous echogenicity [7,29].

Regarding cystic areas, these features were present in 46.6% of LMS cases reported by Ludovisi et al., likely due to necrosis [10]. However, some authors have reported that cystic areas are present in 31.3% of typical leiomyomas and in 55.2% of leiomyomas with cystic or myxoid degeneration.

Therefore, in a single benign myometrial mass may coexist suspicious and non-suspicious ultrasound features, making the evaluation complex.

## 5. Conclusions

Ultrasound is a harmless and inexpensive useful tool for assessing myometrial tumors. Unfortunately, the presence of both suspicious and non-suspicious features in the same lesion and the finding of suspicious features in uterine fibroids make this assessment complex.

We presented our single-center experience in ultrasound imaging assessment in LMS, highlighting that the misleading finding of shadowing and absent or minimal vascularization, generally considered benign ultrasound features, may also occur in LMS patients.

Therefore, to avoid pitfalls, it is important to thoroughly evaluate all suspicious and non-suspicious ultrasound features, as well as the possibility of their coexistence, in all preoperative ultrasound evaluations of myometrial tumors.

## Figures and Tables

**Figure 1 diagnostics-13-00543-f001:**
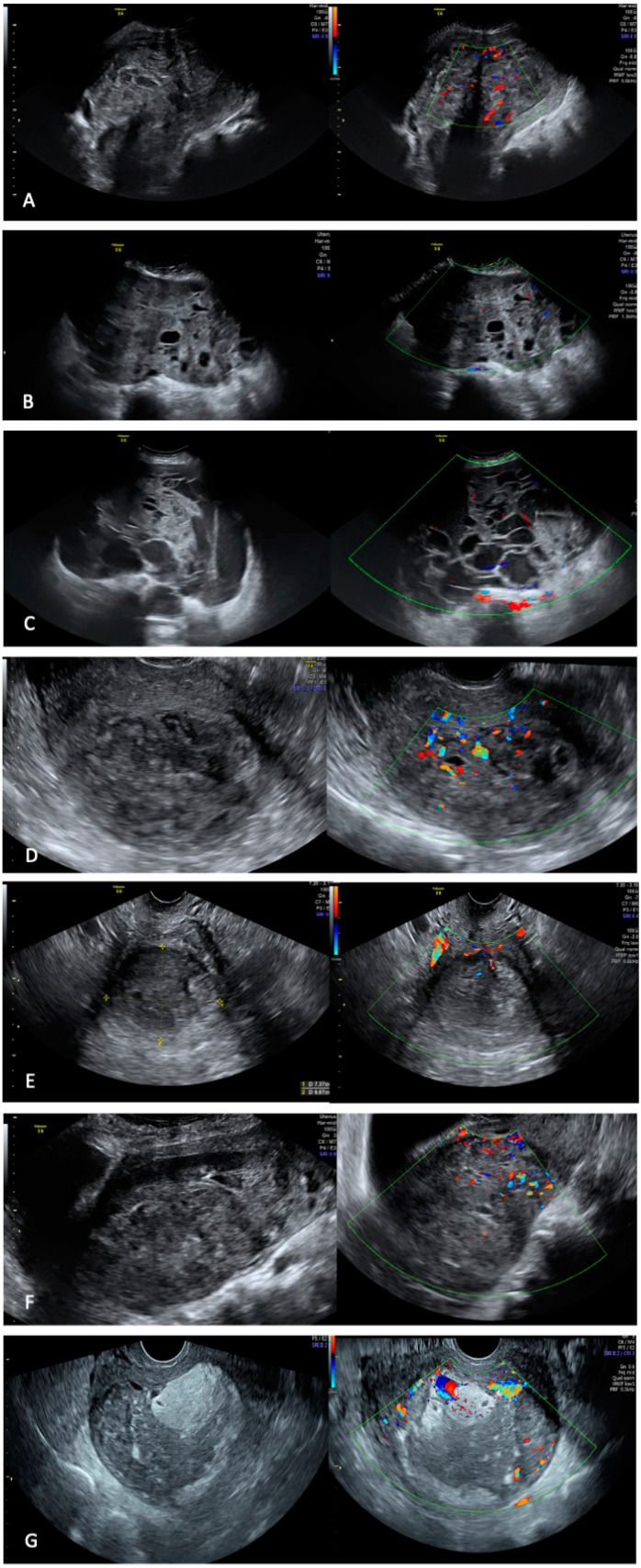
Ultrasound images of LMS patients of our series. Panel (**A**). single myometrial tumor with in-homogeneous echogenicity of the solid tissue, irregular tumor borders, and moderate intralesional vascularization, without cystic areas and with fan-shaped shadowing. The lesion had a soft consistency at probe pressure. Panel (**B**). myometrial tumor with inhomogeneous echogenicity of the solid tissue, irregular tumor borders, and irregular cystic areas with anechoic content, with fan-shaped shadowing. The vascularization was minimal and intralesional. The lesion had a soft consistency under probe pressure. Panel (**C**). single myometrial tumor with in-homogeneous echogenicity of the solid tissue, irregular tumor borders, and irregular cystic areas with low-level content and shadows. The vascularization was minimal and intralesional. The lesion had a soft consistency under probe pressure. Panel (**D**). single myometrial tumor with in-homogeneous echogenicity of the solid tissue, irregular tumor borders, irregular cystic areas with anechoic content, and shadows. The vascularization was moderate and intralesional. The lesion had a soft consistency under probe pressure. Panel (**E**). single myometrial tumor with inhomogeneous echogenicity of the solid tissue, irregular tumor borders, irregular cystic areas with anechoic content, and absence of shadows. The vascularization was minimal and intralesional. The lesion had a soft consistency under probe pressure. Panel (**F**). myometrial tumor with in-homogeneous echogenicity of the solid tissue, irregular tumor borders, irregular cystic areas with low-level content, and absence of shadows. The vascularization was moderate and intralesional. The lesion had a soft consistency under probe pressure. Panel (**G**). single myometrial tumor with inhomogeneous echogenicity of the solid tissue, irregular cystic areas with anechoic content, without shadows, and with minimal intralesional vascularization. The lesion had regular tumor borders and a soft consistency under probe pressure.

**Table 1 diagnostics-13-00543-t001:** Characteristics of LMS patients of our series.

Characteristics	Quantification/Measurement
*Age (mean ± sd)*	50.2 ± 7.0
*Ca125 (mean ± sd)*	28.8 ± 15
*Number of lesions, n (%)*	
Single	5 (71.4%)
Multiple	2 (28.6%)
*Largest diameter (mean ± sd), mm*	152.2 ± 91
*The echogenicity of solid tissue, n (%)*	
Homogeneous	0
Inhomogeneous	7 (100%)
*Tumor borders, n (%)*	
Regular	1 (14.3%)
Irregular	6 (85.7%)
*Shadowing, n (%)* No Yes	3 (42.9%) 4 (57.1%)
*Vascularization, n (%)* Peripheral Intralesional	0 7 (100%)
*Color score, n (%)* 1−2 3−4	4 (57.1%) 3 (42.9%)
*Cystic areas, n (%)* No Yes	1 (14.3%) 6 (85.7%)
*Consistency, n (%)* Hard Soft	0 7 (100%)

## Data Availability

Not applicable.

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
