# Peer review of "Overlap of Suspicious and Non-Suspicious Features in the Ultrasound Evaluations of Leiomyosarcoma: A Single-Center Experience"

_diagnostics, 2023, doi:10.3390/diagnostics13030543_

Round 1
Reviewer 1 Report
The authors attempted to examine the efficacy of ultrasonography to diagnose the uterine leiomyosarcoma. The theme of present study is important; however, there are too many weaknesses in this study. The number of included study is inadequate and lack of control group is critical problem of this study. I’m sorry to say this study provides no new information to the current evidence.
Author Response
Thank you for your review.
The aim of this manuscript is to present our experience with ultrasound evaluation of leiomyosarcoma as a case series rather than a retrospective/case control study.
Reviewer 2 Report
I was glad to read and evaluate the article entitled “Overlap of suspicious and nonsuspicious features in the ultra-sound evaluations of Leiomyosarcoma: a single center experience”, which falls within the aims and scope of the Journal.
In this paper, the authors collected data on a total of seven cases diagnosed with leiomyosarcomas over the period of two years in a single-centre. The methodology used in this manuscript seems adequate, with a proper discussion of the findings. Moreover, the data are supported with quite informative and well explained original images. Although the manuscript can be considered overall of high quality, I would suggest considering the following recommendations for minor revision:
I suggest the authors provide data on how many patients underwent surgery for myometrial tumors in their center. Which was the frequency of LMS in their center over the study period?
Methods: I am reluctant to accept the statement presented in the line 216 “Institutional Review Board Statement: Not applicable.” For this reason, I strongly suggest the authors to request Ethic Committee approval from the institution where patients were examined and operated. Such an approval must be written in the Methods section.
Results: Table 1. should be reorganized for clarity (rows and columns). I would suggest presenting mean±SD in a common manner, without brackets. Bearing in mind that only seven patients were evaluated, I would suggest providing median and range for patients’ age, number and diameter of lesions, as well, in the text. Also, it would be nice to provide data on CA125, as authors discuss the relation between its value and diagnosis of leiomyosarcomas. What about its values in this particular group of patients? There is a reference missing in lines 191-193.
Discussion: It should be reorganized in terms of too many paragraphs in the text. Line 135-136: sentence is unclear. References should be properly inserted in the text.
Conclusions: Line 208-209: please rephrase this segment of the Conclusions in line with the obtained results and statements in the discussion.
References: Citation style throughout the manuscript needs revision. Reference numbers in the text should be inserted in a proper manner. Entire reference list needs revision: the association between ref#25 and topic of the presented manuscript is unclear, as well as the citation context. Ref#24 is not properly cited, i.e., Cho H yon ...
Author Response
I was glad to read and evaluate the article entitled “Overlap of suspicious and nonsuspicious features in the ultra-sound evaluations of Leiomyosarcoma: a single center experience”, which falls within the aims and scope of the Journal.
In this paper, the authors collected data on a total of seven cases diagnosed with leiomyosarcomas over the period of two years in a single-centre. The methodology used in this manuscript seems adequate, with a proper discussion of the findings. Moreover, the data are supported with quite informative and well explained original images. Although the manuscript can be considered overall of high quality, I would suggest considering the following recommendations for minor revision:
I suggest the authors provide data on how many patients underwent surgery for myometrial tumors in their center. Which was the frequency of LMS in their center over the study period?
Thank you for your appreciation.
In the observation period, n. 1000 patients underwent surgery for myometrial pathology. Therefore, the frequency of this disease in our center resulted 0.7%.
Methods: I am reluctant to accept the statement presented in the line 216 “Institutional Review Board Statement: Not applicable.” For this reason, I strongly suggest the authors to request Ethic Committee approval from the institution where patients were examined and operated. Such an approval must be written in the Methods section.
We apologize but it was a mistake. We entered the protocol code in the methods.
Results: Table 1. should be reorganized for clarity (rows and columns). I would suggest presenting mean±SD in a common manner, without brackets. Bearing in mind that only seven patients were evaluated, I would suggest providing median and range for patients’ age, number and diameter of lesions, as well, in the text. Also, it would be nice to provide data on CA125, as authors discuss the relation between its value and diagnosis of leiomyosarcomas. What about its values in this particular group of patients? There is a reference missing in lines 191-193.
We reorganized the table. We modified the presentation of mean±SD. We proved in the text median and range and also reported CA125 values. We added the correct reference.
Discussion: It should be reorganized in terms of too many paragraphs in the text. Line 135-136: sentence is unclear. References should be properly inserted in the text.
We reduced the number of paragraphs, reformulated line 135-136 and checked references.
Conclusions: Line 208-209: please rephrase this segment of the Conclusions in line with the obtained results and statements in the discussion.
We modified the conclusion.
References: Citation style throughout the manuscript needs revision. Reference numbers in the text should be inserted in a proper manner. Entire reference list needs revision: the association between ref#25 and topic of the presented manuscript is unclear, as well as the citation context. Ref#24 is not properly cited, i.e., Cho H yon .
We checked and corrected the citations and their order.
Round 2
Reviewer 1 Report
The authors have made great efforts to improve the manuscript. However, I'm sorry to say that the authors could not address my previous comments. As the authors mentioned that the differential diagnosis is difficult, the reviewer thinks that data from a control group are essential to discuss for the diagnosis of leiomyosarcoma.